# Modelling of Microperforated Panel Absorbers with Circular and Slit Hole Geometries

Pedro Cobo

Institute of Physical and Information Technologies (ITEFI), Spanish National Research Council (CSIC), Serrano 144, 28006 Madrid, Spain; pedro.cobo@csic.es; Tel.: +34-915-618-806

**Abstract:** Although the original proposal of microperforated panels by Maa consisted of an array of minute circular holes evenly distributed in a thin plate, other hole geometries have been recently suggested that provide similar absorption curves to those of circular holes. With the arrival of modern machining technologies, such as 3D printing, panels microperforated with slit-shaped holes are being specially considered. Therefore, models able to predict the absorption performance of microperforated panels with variable hole geometry are needed. The aim of this article is to analyze three models for such absorbing systems, namely, the Maa model for circular holes, the Randeberg–Vigran model for slit-shaped holes, and the Equivalent Fluid model for both geometries. The absorption curves predicted for these three models are compared with the measured curves of three panels microperforated with spirally shaped slits.

**Keywords:** sound absorption; microperforated panels; absorption models

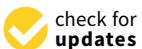



## 1. Introduction

The main advantages of microperforated panels (MPP) as light, clean and easy-to-design sound absorbers were already established in the previous century [1–3]. Although porous and fibrous materials provide satisfactory sound absorption in many noise control applications, they can produce particle discharge, which can discourage their use in hospitals, food industry, white rooms, or in exhaust systems with high flow velocity, such as wind tunnels. In MPPs, the absorption is produced when sound goes through minute holes of diameter $d$ and thickness $t$, due to viscous–thermal losses. The ratio of the open surface to the panel surface is named perforation ratio, or porosity, $\varphi$. Such a distribution of perforations in a panel has acoustic impedance, $Z_1$, which is complex. A good absorber must have impedance matched to that of the air, $Z_0$, which is real. Therefore, in order that such an MPP provides enough absorption, other complex impedance is needed to compensate for the reactive part of $Z_1$. This can be carried out by adding an air cavity of thickness $D$ in front of the perforated panel. Thus, an MPP consists of a parametric, tunable absorber which depends on $(d,t,\varphi,D)$ [4].

If these parameters are properly tuned, the MPP is able to provide a clean absorber with a frequency band of 1–2 octaves [5]. The bandwidth of a single-layer MPP can even be increased to more than two octaves at medium-high frequencies using minute holes ($d < 0.15$ mm) with a high perforation ratio ($\varphi > 20\%$) [6]. Other ways to increase the absorption band include designing multiple-layer MPPs [7–10], incompletely partitioning the back cavity [11], or backing the MPP with a porous layer [12,13].

Conventional MPPs are known to provide limited absorption at low frequencies (below 500 Hz). However, special structures have been recently proposed for increasing the low-frequency absorption of MPPs, including parallel arrays of MPPs [14,15], MPPs backed by Helmholtz resonators [16], or honeycombs backed by MPPs [17].

The first model to obtain the absorption curve of an MPP based on parameters $(d,t,\varphi,D)$ was provided by Maa [1,2], who combined the Crandall solution of the sound inside tubes with the edge effects identified by Ingard [18]. More recently, an equivalent fluid (EF)

model has been proposed for MPPs [19,20]. Both models afford very similar absorption curves for an MPP with the same set of parameters ($d,t,\varphi,D$).

An MPP appropriate for absorbing in the frequency band of interest in noise-control applications requires hundreds of thousands of sub-millimetric perforations per square meter. Usually, such large numbers of minute perforations would be carried out by laser technology, resulting in an expensive absorber. There have been proposals to manufacture cheaper MPPs, such as microslit panels (MSP), in which slit-shaped perforations, rather than circular holes, are created [21–25]. Recently, 3D printing has been suggested for machining MSPs [26,27], providing cheap and easy-to-design absorbers with sound absorption performance very similar to that of MPPs. However, while Maa's and the EF models provide theoretical predictions which match reasonably well with experimental data for MPPs, experimental validation of MSPs has thrown up predicted–measured disagreements [27]. Therefore, the main objective of this article is to revisit the modelling of MPPs and MSPs and compare them with updated measured absorption data.

## 2. Single-Layer Microperforated or Microslit Panels

A single-layer MPP or MSP consists of a plate of thickness $t$, with perforations of hydraulic diameter $d$, in front of an impervious wall, leaving an empty cavity of thickness $D$. These perforations can be made of arbitrary geometry, including circular holes or rectangular slits. The percentage of plate open surface is named perforation ratio, or porosity, and is denoted by $\varphi$. Such a system is characterized by an input impedance, $Z_1$. When a plane wave, propagating in air, with characteristic impedance $Z_0$, reaches the MPP or MSP, the impedance contrast ($Z_1 - Z_0$) causes a reflection, and hence, an absorption. At normal incidence, the reflection, $R$, and absorption, $\alpha_0$, coefficients are

$$R = \frac{Z_1 - Z_0}{Z_1 + Z_0} \tag{1}$$

$$\alpha_0 = 1 - |R|^2 \tag{2}$$

The input impedance, $Z_1$, has four effects [28]:

- viscous–thermal dissipation within the perforations, characterized by impedance $Z_{hole}$ for an MPP, and $Z_{slit}$ for an MSP;
- flow distortion in the perforation edges, characterized by impedance $Z_{edge}$;
- resonances in the air cavity, with impedance $Z_D$;
- structural vibrations of the panel impinged by the incident acoustic field, with impedance $Z_{vib}$.

The contribution of $Z_{hole}$ and $Z_{edge}$ is named $Z_{MPP}$ for an MPP (or $Z_{MSP}$ for an MSP). The air cavity impedance is:

$$Z_D = -iZ_0 cot(kD) \tag{3}$$

where $k = \omega/c$ is the wavenumber, $\omega = 2\pi f$ is the angular frequency, $c$ is the sound velocity in air, and $f$ is the frequency. The structural impedance of the panel, $Z_{vib}$, can be obtained as a function of the elastic parameters of the panel. The input impedance, $Z_1$, can be obtained as [29]:

$$Z_1 = \frac{Z_{MPP}Z_{vib}}{Z_{MPP} + Z_{vib}} + Z_D \tag{4a}$$

For an impervious panel $Z_{vib} \rightarrow \infty$ and

$$Z_1 = Z_{MPP} + Z_D \tag{4b}$$

For an MSP, $Z_{MPP}$ in Equation (4b) must be changed for $Z_{MSP}$. In the following, equations for $Z_{MPP}$ and $Z_{MSP}$ will be analyzed.

### 2.1. Maa Model

The hole impedance for circular holes is [1,2]

$$Z_{hole} = i\omega\rho_0 t \left[1 - \frac{2}{s\sqrt{-j}}\frac{J_1\left(s\sqrt{-i}\right)}{J_0\left(s\sqrt{-i}\right)}\right]^{-1} \tag{5}$$

where

$$s = d\sqrt{\frac{\rho_0\omega}{4\eta}} = r\sqrt{\frac{\rho_0\omega}{\eta}} \tag{6}$$

and $\eta$ is the dynamic air viscosity; $s$ represents the ratio of the hole diameter to the viscous boundary layer thickness. To extrapolate the impedance at the tube input to the impedance of an MPP (an array of parallel tubes) we need to take into account the relation between the particle velocities within ($u$) and outside ($u'$) the perforations

$$u = \frac{u'}{\phi} \tag{7}$$

Therefore, the hole impedance for an array of circular perforations is:

$$Z_{hole} = i\frac{\omega\rho_0}{\phi}t\left[1 - \frac{2}{s\sqrt{-j}}\frac{J_1\left(s\sqrt{-i}\right)}{J_0\left(s\sqrt{-i}\right)}\right]^{-1} \tag{8}$$

Strictly speaking, Maa used an approximation of Equation (8) in the range $1 < s < 10$ for $Z_{hole}$. However, the exact Equation (8) will be used in the following.

For the edge impedance of circular holes, Maa [1,2] proposed adding two terms, one resistive, due to the friction of air flow in the openings of the holes, and other reactive, due to the sound radiation of the two openings. The resistive term, $R_s$, is also named the surface resistance or resistance end correction, and the reactive term, $X_m$, is called the mass reactance or reactance end correction.

The resistance end correction is due to air flow friction on the surface of the panel, as the air flow is squeezed into the small area of the inlet end of the perforation [30]. For the resistive term, Ingard assumed a flow distortion covering an area including a semi-sphere of radius $r$ centered at the hole [18]. Therefore, for circular holes:

$$R_m = \frac{\sqrt{2\eta\omega\rho_0}}{2} \tag{9}$$

For the reactive part, it a vibrant mass of length $d + 2\delta$, larger than the panel thickness, is assumed. The mass reactance end correction, which is calculated from the sound radiation impedance of a piston of the same shape, is important for these short tubes. For the mass reactance end correction of the narrow slit cross section, Maa utilized the radiation impedance of an elliptical piston [30]. Therefore, $X_m = i\rho_0\omega(d + 2\delta)$. For circular holes, $\delta = 0.48\sqrt{A}$, $A$ being the area of the hole surface, which is $A = \pi r^2$. Thus:

$$X_m = i\rho_0\omega\, 0.85\, d \tag{10}$$

So the edge impedance of a circular hole is:

$$Z_{edge} = R_s + iX_m = \frac{\sqrt{2\eta\omega\rho_0}}{2} + i\omega\rho_o 0.85d \tag{11a}$$

which can also be written as a function of $s$ (extrapolating also to a panel of perforation ratio $\varphi$):

$$Z_{edge} = \frac{\sqrt{2}\eta s}{\phi d} + i\frac{\omega\rho_o 0.85d}{\phi} \tag{11b}$$

This edge impedances assumes that the perforations are far enough from each other (low perforation ratio) that there are no interactions between holes. However, for high perforation ratios, the edge interaction effect has to be taken into account. For circular holes, Melling [31] proposed introducing this effect in the mass reactance, $X_m$, through:

$$X_m = i\omega\rho_0 0.85\ d\ F_1(\epsilon) \tag{12}$$

where:

$$F_1(\epsilon) = 1 - 1.4092\ \epsilon + 0.33818\ \epsilon^3 + 0.06793\ \epsilon^5 - 0.02287\ \epsilon^6 + 0.03015\epsilon^7 - 0.01641\epsilon^8 \tag{13}$$

where the Fok function [32] is $\epsilon = \sqrt{\phi}$.

Thus, the equation for the input impedance of an MPP according to the Maa model is:

$$Z_{1,Maa} = \frac{\sqrt{2}\eta s}{\phi d} + i\frac{\omega\rho_0}{\phi}\left\{0.85\ d\ F_1(\epsilon) + t\left[1 - \frac{2}{s\sqrt{-i}}\frac{J_1\left(s\sqrt{-i}\right)}{J_0\left(s\sqrt{-i}\right)}\right]^{-1}\right\} - iZ_0 cot(kD) \tag{14}$$

where $F_1$ is the overperforation correction factor of Equation (13).

### 2.2. Randeberg–Vigran (RV) Model

For a panel microperforated with rectangular slits (MSP) of hydraulic diameter $d$, the slit impedance is [23,24]:

$$Z_{slit} = i\frac{\omega\rho_0}{\phi}t\left[1 - \frac{tanh\left(s\sqrt{i}\right)}{s\sqrt{i}}\right]^{-1} \tag{15}$$

Vigran and Haugen proposed the following edge impedance for slits of hydraulic diameter $d$ and length $l$, [25]:

$$Z_{edge} = i\ 0.636\ \frac{\rho_0\omega d}{\phi}\left\{\frac{\beta}{3} + \frac{1 - (1+\beta^2)^{3/2}}{3\beta^2} + \left[\frac{ln\left(\beta + \sqrt{1+\beta^2}\right)}{\beta} + ln\left(\frac{1 + \sqrt{1+\beta^2}}{\beta}\right)\right]\right\} \tag{16a}$$

where $\beta = d/l$. For low $\beta$ (low width to length ratio), the following approximate formula can be used [10]:

$$Z_{edge} = -i\ \frac{2}{\pi}\ \frac{\rho_0\omega d}{\phi}ln\left[sin\left(\frac{\phi\pi}{2}\right)\right] \tag{16b}$$

Notice that the Vigran's end correction is purely reactive. Although the resistive edge correction is negligible for slits, Kristiansen and Vigran [33] also included a resistive term in the edge impedance of the MSP for reasons of completeness. With this term, this impedance is:

$$Z_{edge} = \frac{\sqrt{2}\ \rho_0\omega\eta}{2\phi} - i\ \frac{2}{\pi}\ \frac{\rho_0\omega d}{\phi}ln\left[sin\left(\frac{\phi\pi}{2}\right)\right] \tag{16c}$$

Chevillotte [34] proposed a correction factor which is valid for plates with squared slits, to take into account the interaction effects in overperforated MSPs

$$F_2(\epsilon) = 1 - 1.25\ \epsilon \tag{17}$$

and Vigran provided the following equation for the correction factor for slits [24]

$$F_3(\epsilon) = 1 - 1.25\,\epsilon + 0.47\,\epsilon^3 \qquad (18)$$

Figure 1 shows the inter-comparison of the above correction factors, including the correction for holes (Equation (13)). As it can be seen, all the functions tend to 1 for low perforation ratios and tend to 0 for high perforation ratios. Therefore, the overperforation correction factor tends to shorten the excess length of the air cylinder vibrating within the holes. The correction factor for slits decreases less rapidly than that for circles. The correction factor for squares is roughly in between of the corrections factors for circles and slits.

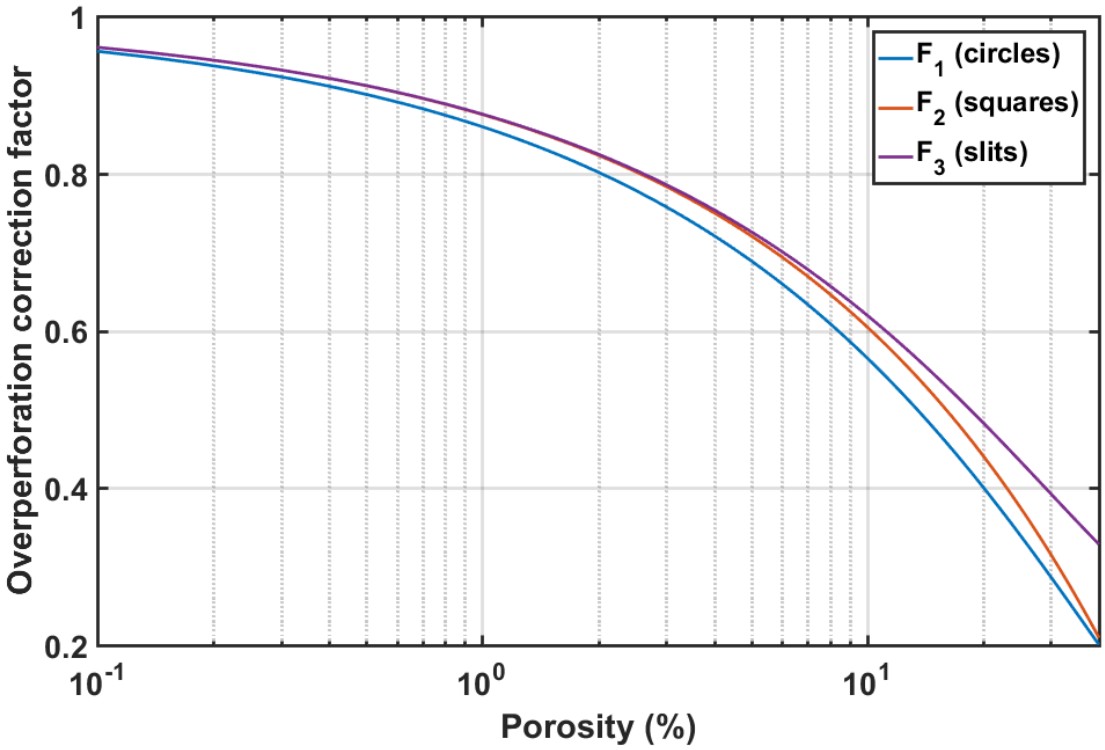

**Figure 1.** Comparison of different correction factors for overperforation.

Jiang et al. [35] used an additional length model to describe the reactive part of the impedance end correction of microperforated panels, which is extended to describe the resistive part. The cross-sectional impedance is computed along the axis of one perforation cell with a circular hole. Except for the obvious jumps in the narrow regions at the inlet and outlet of the perforation, the impedance varies linearly along the axis following exactly that of the viscous wave in the circular hole. The additional length for the impedance end correction is obtained by extrapolating the linearly varying impedance inside the hole. Empirical models for the resistive and reactive additional lengths have been obtained based on viscous–thermal acoustic simulation with 96 test cases and can be found in [35].

Therefore, the complete RV model for the input impedance of an MSP is:

$$Z_{1,RV} = \frac{\sqrt{2\,\rho_0 \omega \eta}}{2\phi} + i\frac{\omega \rho_0}{\phi}\left\{ t\left[1 - \frac{tanh\left(s\sqrt{i}\right)}{s\sqrt{i}}\right]^{-1} - 0.636\,d\,F_3\left(\sqrt{\phi}\right)\,ln\left[sin\left(\frac{\phi\pi}{2}\right)\right]\right\} - iZ_0 cot(kD) \qquad (19)$$

where $F_3$ is the overperforation correction factor for slits in Equation (23).

### 2.3. Equivalent Fluid (F) Model

An equivalent fluid (EF) form of MPP, based on the Johnson-Champoux-Allard model for porous absorbers, has been proposed [19,20]. A perforated plate of infinite lateral extension is assumed, coupled at both sides to a semi-infinite fluid. The impedance of such a plate has a resistive and a reactive part. The resistive part is induced by viscous effects within the viscous boundary layer and around the edges of the panel holes, due to the distortion of the acoustic flow. The reactive part takes into account the movement of the air mass in the perforation, which is larger than the panel thickness. Both the charge of the mass associated with the sound radiation of hole openings and the distortion of the acoustic flow at the panel surface contribute to making the hole neck heavier and more difficult to move.

Allard and Atalla [20] proposed the following equations for the normalized effective density, $\rho$, and bulk modulus, $K$, of perforated panels:

$$\frac{\rho}{\rho_0} = 1 + \frac{\sigma\phi G_c(s)}{i\rho_0\omega} \tag{20}$$

and

$$\frac{K}{P_0} = \frac{\gamma}{\gamma - (\gamma - 1)F(B^2 s)} \tag{21}$$

where

$$G_c(s) = \frac{-\frac{s\sqrt{-i}}{4}\frac{J_1(s\sqrt{-i})}{J_0(s\sqrt{-i})}}{1 - \frac{2}{s\sqrt{-i}}\frac{J_1(s\sqrt{-i})}{J_0(s\sqrt{-i})}} \tag{22}$$

$$F(B^2 s) = \frac{1}{1 + \frac{\sigma\phi}{iB^2\rho_0\omega}G_c(Bs)} \tag{23}$$

and $\gamma$ is the ratio between the specific heats of air at pressure and volume constant, $P_0$ is the ambient pressure, $B^2$ is the Prandtl number, and $\sigma$ is the flow resistivity. For air at 18 °C, $\gamma = 1.4$, $K_0 = 1.0132\ 10^5$ Pa, and $B^2 = 0.71$. The hole geometry is introduced in this formulation through two constants, $C_1$ and $C_2$ [20]:

$$s = C_1 r \sqrt{\frac{\rho_0\omega}{\eta}} \tag{24a}$$

$$\sigma = C_2 \frac{\eta}{\phi r^2} \tag{24b}$$

Table 1 summarizes the values of the constants $C_1$ and $C_2$ for different cross-sectional hole geometries [20].

**Table 1.** Values of the constants $C_1$ and $C_2$ for different cross-sectional hole geometries.

|       | Circle | Square | Equilateral Triangle | Slit |
|-------|--------|--------|----------------------|------|
| $C_1$ | 1      | 1.07   | 1.11                 | 0.81 |
| $C_2$ | 8      | 7      | 6.5                  | 12   |

Given the effective density and bulk modulus of the equivalent fluid of the microperforated plate, the characteristic impedance, $Z_c$, and propagation constant, $k_c$, can be obtained from the equations

$$Z_c = \sqrt{K\rho} \tag{25}$$

$$k_c = \omega\sqrt{\frac{\rho}{K}} \tag{26}$$

A similar EF formulation for MPPs, but disregarding thermal effects, was proposed by Atalla and Sgard [19].

The parametric *EF* model outlined in Equations (20)–(26) can be used for both MPPs and MSPs. The complete *EF* model requires adding the impedance of the air layer, of depth *D*, to obtain:

$$Z_{1,EF} = \frac{Z_c}{\phi} \frac{Z_D cos(k_c t) + i \frac{Z_c}{\phi} sin(k_c t)}{\frac{Z_c}{\phi} cos(k_c t) + i Z_D sin(k_c t)} \tag{27}$$

where $Z_c$ and $k_c$ are the characteristic impedance and the propagation constant, respectively, of the equivalent fluid of the perforated panel given by Equations (25) and (26), respectively.

### 2.4. Comparison of Models

The performance of these models can be illustrated by comparing the absorption curves resulting from different combinations of parameters. For these comparisons, the MSPs are assumed to have the same hydraulic diameter and perforation ratio as the corresponding MPPs. Figure 2 shows the results for four combinations of the parameters $(d,t,\varphi,D)$.

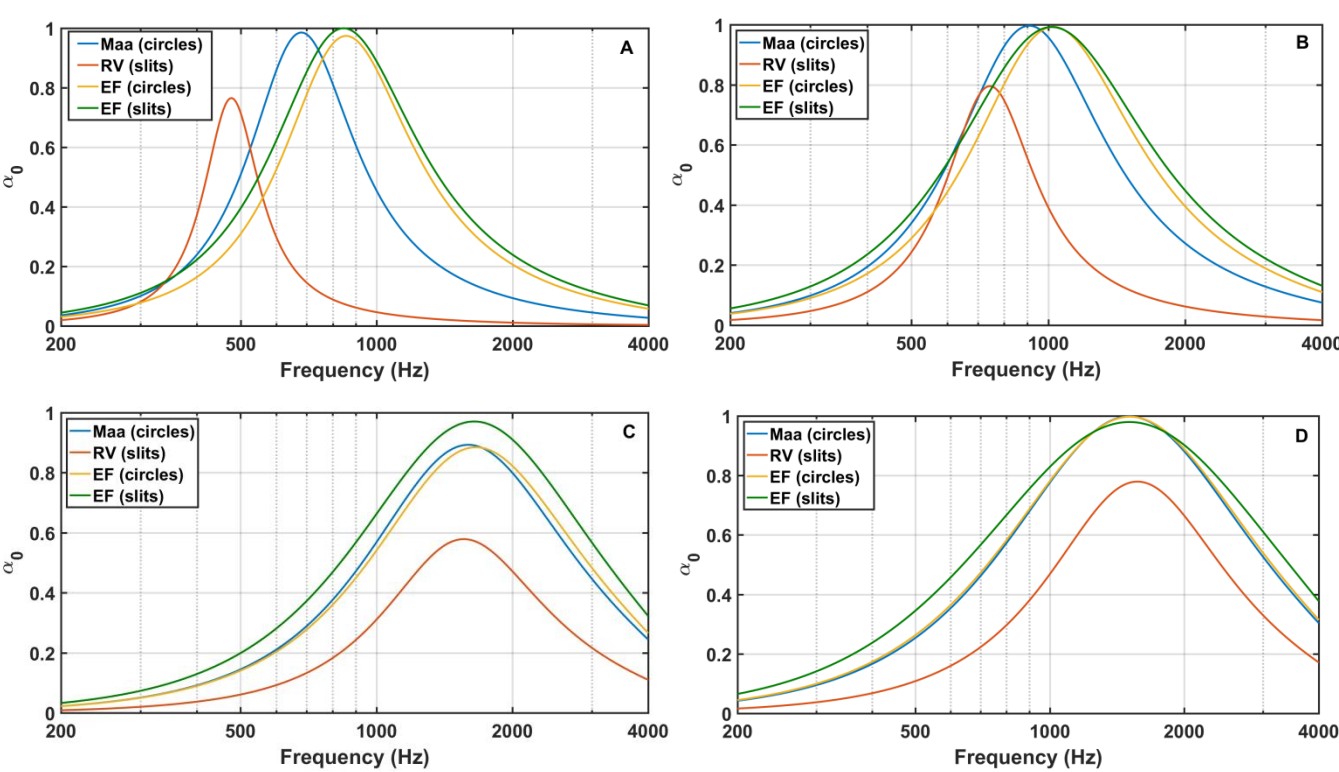

**Figure 2.** Absorption curves of MPPs and MSPs with varoius combinations of parameters: (**A**) $(d,t,\varphi,D)$ = (0.5 mm, 0.5 mm, 0.5%, 3 cm); (**B**) $(d,t,\varphi,D)$ = (0.35 mm, 0.65 mm, 1%, 3 cm); (**C**) $(d,t,\varphi,D)$ = (0.25 mm, 1 mm, 5%, 3 cm); (**D**) $(d,t,\varphi,D)$ = (0.2 mm, 5 mm, 20%, 3 cm).

Comparing these curves, the following observations can be made:

- When the perforation ratio increases (overperforation) and the perforation diameter decreases, the absorption peak moves towards higher frequencies and the absorption bandwidth becomes wider [6].
- It seems that the Maa model of MPP always provides more absorption than the RV model for the equivalent MSP. The absorption peaks may be displaced towards lower frequencies for low perforation ratios, although the bandwidth of the Maa curve is wider than that of the RV model for the equivalent MSP, for the considered combinations of parameters.

- There are discrepancies between the absorption curves provided by the Maa model and the EF model for circular holes at low perforation ratios (Figure 2A,B). In this case, the EF curve is slightly displaced towards higher frequencies with respect to the Maa curve. However, when the perforation ratio increases (Figure 2C,D), both curves tend to match each other.
- The discrepancies between the absorption curves provided by the RV and the EF models for slits are large for all combinations of parameters. For low perforation ratios (Figure 2A,B) the EF curve is displaced towards higher frequencies. For high perforation ratios (Figure 2C,D) the EF curve has a higher peak and wider bandwidth.

### 3. Results

For practical reasons, it may be convenient to design MSPs with spiral-shaped slits, which can be more aesthetical and easier to machine. For these MSPs (Figure 3) we can use the equations of the spiral to calculate the porosity. The equation in polar coordinates $(r,\theta)$ for such a spiral is:

$$r = \frac{Q}{2\pi} \cos \theta \tag{28}$$

where $Q$ is the incremental distance between spirals. The spiral length, $L$, is given by the equation:

$$L = \frac{\pi}{Q} \left[ NQ \sqrt{\left(\frac{Q}{2\pi}\right)^2 + (NQ)^2} + \left(\frac{Q}{2\pi}\right)^2 \left\{ ln \left| NQ + \sqrt{\left(\frac{Q}{2\pi}\right)^2 + (NQ)^2} \right| - ln\left(\frac{Q}{2\pi}\right) \right\} \right] \tag{29}$$

$N$ being the number of turns. If the panel is perforated with spirals of hydraulic diameter $d$, the open surface will be $Ld$, and the perforation ratio is:

$$\phi = \frac{Ld}{\pi R^2} \tag{30}$$

where $R$ is the panel radius (usually, the inner radius of the impedance tube where the absorption coefficient will be measured).

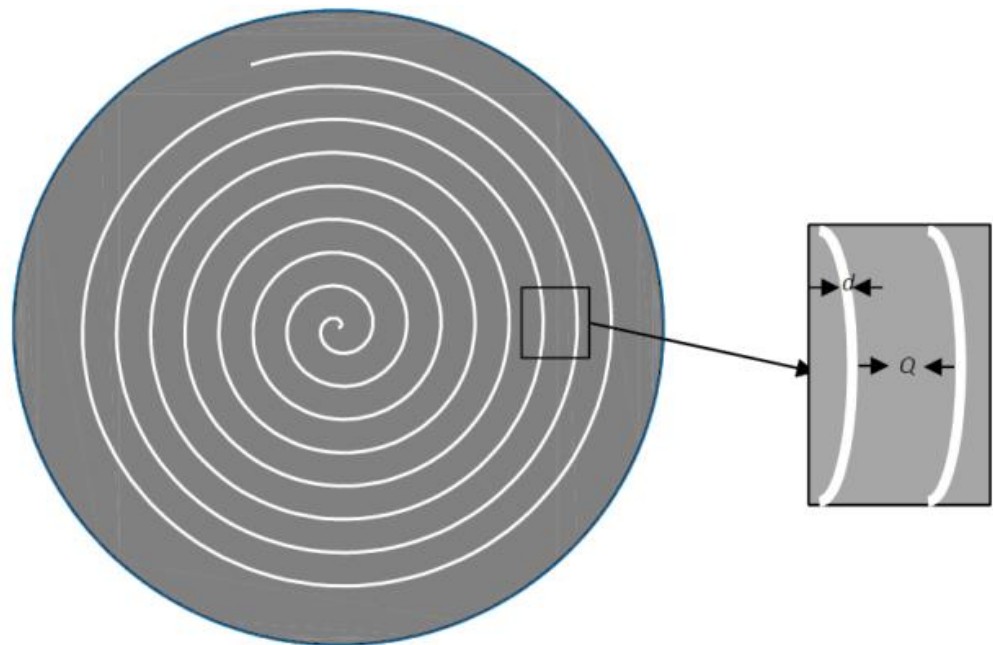

**Figure 3.** Sketch of a spiral-slotted MSP.

Three MSPs with spiral-shaped slits, with fixed $(t,D) = (6\text{ mm}, 3\text{ cm})$ and variable $(d,\varphi)$ were machined by 3D printing to be measured in a circular tube of inner diameter of 30 mm ($R_1 = 15$ mm). Slotted panels were manufactured by successive pouring of hot PolyLactic Acid (PLA) bioplastic. Table 2 summarizes the combination of the nominal parameters of these MSPs. The three MSPs were overperforated. The first one, MSP8, had 6 turns of a slit 0.44 mm wide, resulting in a perforation ratio of 16%. The second MSP, MSP9, had 7 turns of a slit of hydraulic diameter 0.44 mm. Its perforation ratio was 19%. The slit of the third MSP, MSP11, was 0.35 mm wide and was machined in a spiral of 8 turns, resulting in a porosity of 18%. The normal incidence absorption coefficient of these MSPs was measured in an impedance tube by means of the Transfer Function method [36]. The transfer function between two $\frac{1}{4}$ inch condenser microphones separated by 50 mm was used to calculate the normal incidence sound absorption coefficient of the three MSPs with spiral-shaped slits, mounted at one end of an impedance tube with an inner diameter of 30 mm, in the frequency range of 200 to 4000 Hz. Special care was taken to ensure the linearity of microphone measurements in the tube. These absorption curves are shown in Figure 4. They have peaks at 1200 Hz (0.88), 1440 Hz (0.65), and 1350 Hz (0.81), respectively.

**Table 2.** Parameters of the spirally slotted MSPs with fixed $(t,D) = (6\text{ mm}, 3\text{ cm})$.

| MSP | $d$ (mm) | $N$ | $\varphi$ (%) |
|---|---|---|---|
| MSP8  | 0.44 | 6 | 16 |
| MSP9  | 0.44 | 7 | 19 |
| MSP11  | 0.35 | 8 | 18 |

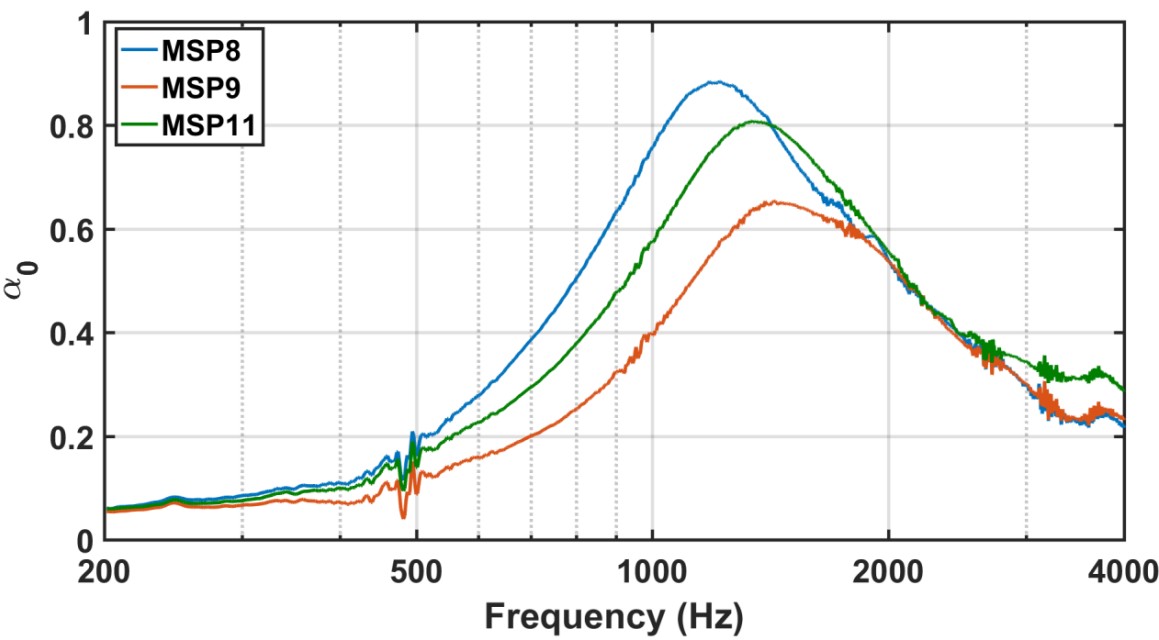

**Figure 4.** Measured sound absorption curves of MSP8, MSP9 and MSP11.

Figures 5–7 show the comparison of the measured absorption curves of these MSPs with the theoretical absorption curves provided by the three models (in the case of the EF model, the couple of parameters for slits (C1, C2) = (0.81, 12) were used). As can be seen, none of the absorption curves provided by these models for the combination of nominal parameters shown in Table 2 closely match the measured absorption curves [37]. Nevertheless, the model in which the absorption curves approach the measured curves best is the Maa model, even though this model was originally proposed for circular holes. The absorption curves provided by the EF model are closer to the measured curves than those of the RV model.

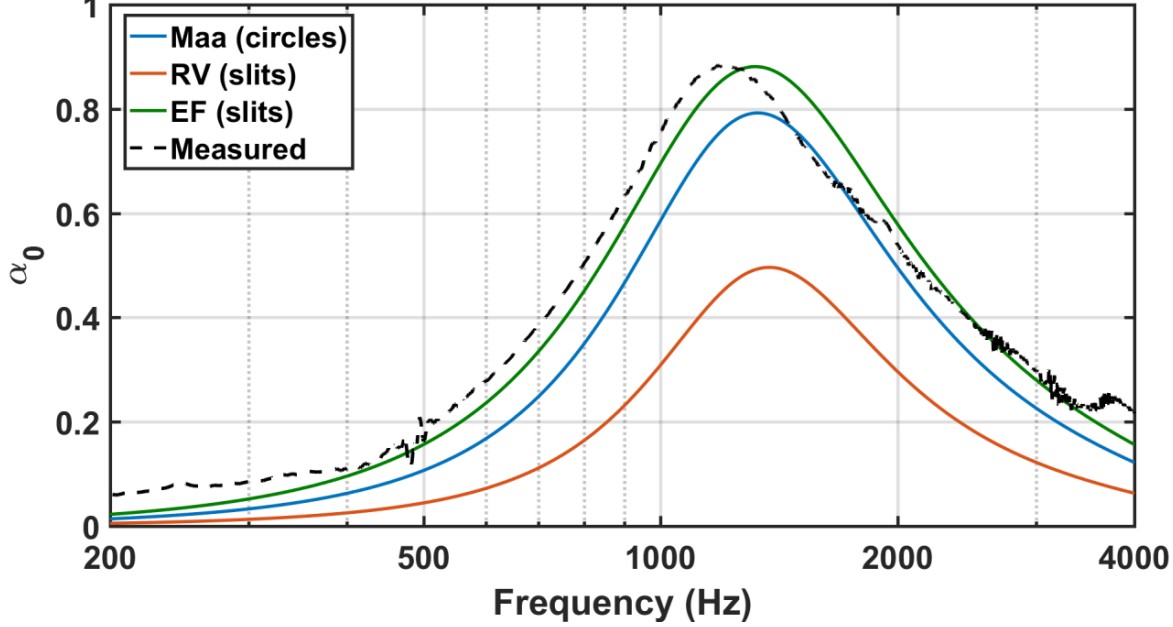

**Figure 5.** Comparison of the measured sound absorption curve of MSP8 with those provided by the three models.

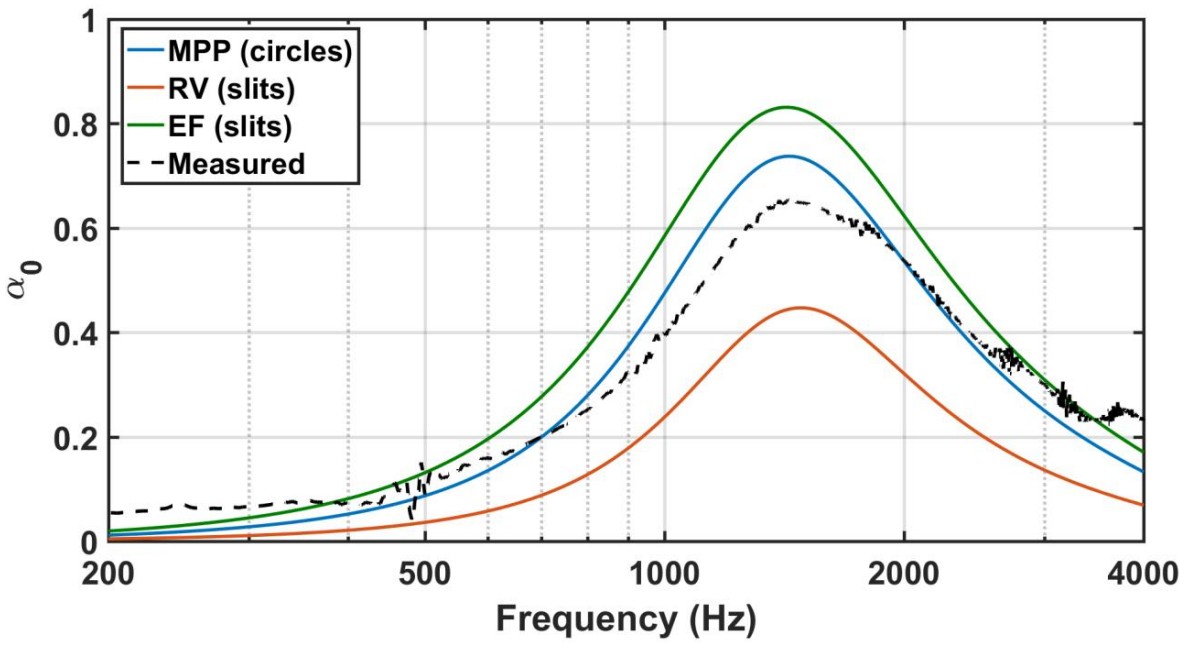

**Figure 6.** Comparison of the measured sound absorption curve of MSP9 with those provided by the three models.

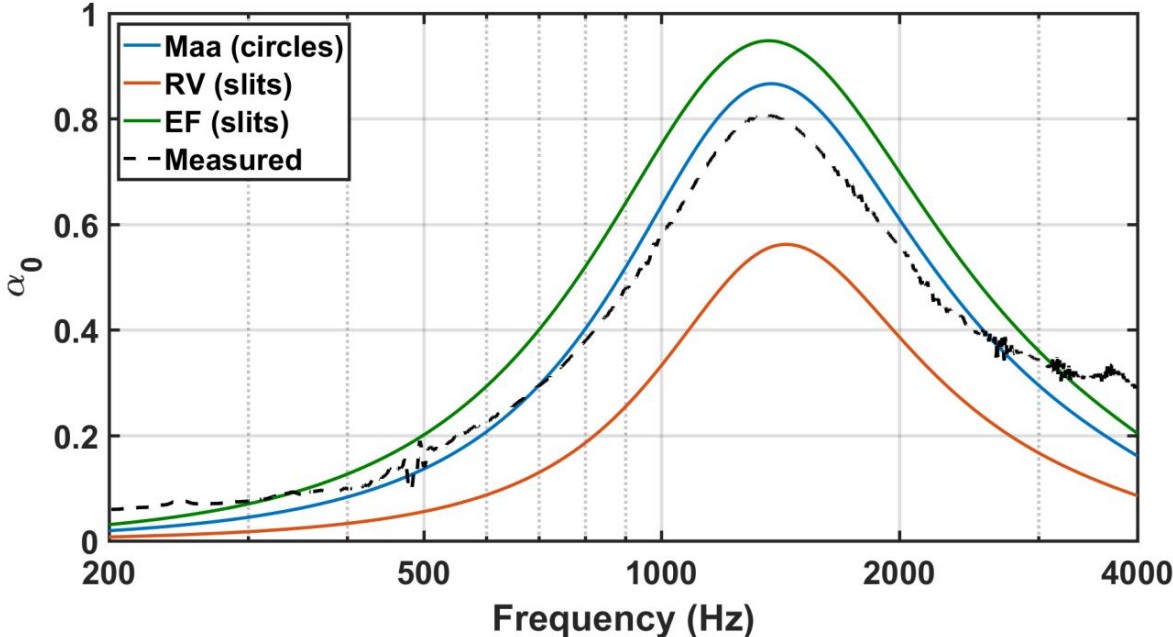

**Figure 7.** Comparison of the measured sound absorption curve of MSP11 with those provided by the three models.

### 4. Discussion

Although analytical equations were used for the three models discussed above, some of their terms, especially the end correction term, are of numerical/empirical origin. Analytical equations could be used to calculate the edge impedance in the perforations using equations for the pressure and velocity fields around the pores. However, such equations are so complicated, even for simple pore geometries, that they can only be approximately solved by numerical methods. For instance, Li [38] used the viscothermal wave theory to present an analytical solution for the end correction of sharp-edged circular holes in MPPs. An impedance end correction model is then derived from the asymptotic expansion of the modal solution. Naderyan et al. [39] combined analytical solution for perforated plates with Finite Element Method (FEM) to develop numerical formulas for the reactive and

resistive end effects of the perforations on the plate. Aulitto et al. [40], on the other hand, used numerical solutions to the Navier-Stokes equation to provide expressions for the end-corrections of microslit absorbers as functions of the ratio of the slit height to viscous boundary layer thickness (shear number) and the porosity.

While these approaches assume perfect perforations, real perforations are not perfect. Ren et al. [41] considered cylindrical perforations with cross-sectional shapes systematically altered around circle and applied numerical analyses based on the viscous–thermal coupled acoustical equations to investigate the tunable acoustic performance of the proposed absorbers and to reveal the underlying physical mechanisms. They demonstrated that pore morphology significantly affects the sound absorption of MPPs by modifying the velocity field (and hence viscous dissipation) in the pores. According to them, the unevenness of pores in MPPs increases the absorption performance, both in the peak and bandwidth of the curves. Xu et al. [42] analyzed the sound absorption of an MPP with petal-shaped perforations. Compared to circular perforations, the petal-shaped perforations decreased the resonant frequency and increased the maximum absorption coefficient of the MPP.

Slit walls machined by 3D printing procedures are expected to be rough. Slotted panels are manufactured by successive pourings of hot bioplastic which is likely to create rather irregular slit walls. Xu et al. [43] used also full numerical simulations with FEM to account for the effect of roughness on the surface of perforations in the sound absorption of MPPs. They found that wall roughness of pores can have a noticeable effect on the absorption performance of the MPP. The presence of surface roughness decreased the resonant frequency and increased the peak absorption coefficient of the analyzed MPPs [43].

The effect of the hole geometry in the above-described EF model is introduced through two constants ($C_1$,$C_2$) affecting the parameter $s$ and the flow resistivity, $\sigma$ (see Table 1). Looking at Figures 5–7, it seems that discrepancies between the EF model's predicted and measured absorption curves are not too large. Therefore, the predictions of the above-described EF model could be improved by the empirical fitting of ($C_1$,$C_2$).

## 5. Conclusions

Three models were analyzed to predict the absorption curves of microperforated plates with circular (MPPs) and slit-shaped (MSPs) holes. Two of these models were originally conceived for circles (Maa model) and slits (Randeberg–Vigran model), while the other (EF model) contains two parameters that allow for its use in both pore geometries. When the absorption curves predicted for these models were compared with the measured ones for MSPs with spiral-shaped slits, a closer agreement was obtained for the Maa model. The parametric EF model provided absorption curves which were not too far from those of the measured curves. Moreover, the agreement between the predicted and measured curves could be improved by empirically fitting the couple of parameters ($C_1$,$C_2$) with spirally machined MSPs of different ($d$,$t$,$\varphi$) values.

**Funding:** This research received no external funding.

**Data Availability Statement:** This study did not report any data.

**Acknowledgments:** F. Simón and Marco Cortés are acknowledged for the machining of MSPs by 3D printing.

**Conflicts of Interest:** The authors declare no conflict of interest.

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
