# Peer review of "Modelling of Microperforated Panel Absorbers with Circular and Slit Hole Geometries"

_acoustics, doi:10.3390/acoustics3040042_

Round 1

Reviewer 1 Report

Improve references eg.
W. Yang. 3D Printing of Polymeric Multi-Layer Micro-Perforated Panels for Tunable Wideband Sound Absorption. Polymers.

The authors propose a spiral system for sound absorption and compare the measured acoustic absorption values with the theoretical values obtained for the MPPs.

I think that the models proposed are for MPP and cannot be adapted to the model under test.

Furthermore, a 3D drawing of the specimen would be better to understand the geometry of the system.

You could apply a calculation method such as:
G. Ciaburro et al. Numerical Simulation for the Sound Absorption Properties of Ceramic Resonators. Fibers

Authors should use computational models more responsive to the chosen geometry.

It is not described how the measurement of the absorption coefficient (kundt tube?) Is performed.

Reviewer 2 Report

I find the research developed in the manuscript interesting.
In this sense, I recommend its publication in the journal with minor changes.
The author should carefully review and correct some misprints such as Line 10 "whit" Line 124 "end" Line 182 "and" does not have to be italic

Reviewer 3 Report

The paper presents and discusses the three models for the prediction of absorption curves of microperforated plates with circular (MPPs) and slit-shaped (MSPs) holes. Although the MAA and EF models provide theoretical predictions that match reasonably well with experimental data for MSPs, there is some disagreement for experimental validation of MSPs. The major focus of this paper was therefore to revisit the modeling aspects of MPPs and MSPs and their relevant comparison with absorption data. The paper presents the Maa model for circular holes, the Randaberg-Vigran model for slit-shaped holes, and the Equivalent Fluid model for both geometries.

In review, I have some suggestions for the authors to implement as follows:

  1. There are several typos scattered throughout the manuscript – Abstract, line 3, whit should be with. The author needs to carefully read the manuscript before submitting the revised version.
  2. The Introduction section is very concise and the discussion on the state-of-the-art is marginal. 
  3. The discussion on the three models needs to be extended from a theoretical perspective as it is the backbone of the present manuscript.
  4. The MSPs with spirally shaped slits machined using 3D machining need some concise information on the process and the material used for the MSPs.
  5. The legend in Fig.2 needs to add to each individual figure.
  6. Are the absorption curves of MPPs and MSPs with different combinations of parameters normalized? If yes, how were they normalized?

Round 2

Reviewer 1 Report

In Figure 2 and Figures 5 to 7,  it would be advisable to start from 200 Hz, not 0 Hz the low-frequency absorption coefficient is small, but the measurement is very difficult.
Row 246, explain better how you performed the measurements of the absorption coefficient with impedance tube (describe the type of tube, microphones, and relative distances).

Reviewer 3 Report

The author has incorporated the suggestions proposed by the reviewers and the manuscript has been revised in accordance. I, therefore, recommend the manuscript for publication in its present form.
